# NLIR: Natural Language Intermediate Representation for Mechanized Theorem Proving

**Laetitia Teodorescu**
Adaptive ML

**Guillaume Baudart**
Université Paris Cité
CNRS, Inria, IRIF

**Emilio Jesús Gallego Arias**
Université Paris Cité
CNRS, Inria, IRIF

**Marc Lelarge**
DI ENS, PSL University
Inria

## Abstract

Formal theorem proving is challenging for humans as well as for machines. Thanks to recent advances in LLM capabilities, we believe natural language can serve as a universal interface for reasoning about formal proofs. In this paper, 1) we introduce *Pétanque*, a new lightweight environment to interact with the Coq theorem prover; 2) we present two interactive proof protocols leveraging natural language as an intermediate representation for designing proof steps; 3) we implement beam search over these interaction protocols, using natural language to rerank proof candidates; and 4) we use Pétanque to benchmark our search algorithms. Using our method with GPT-4o we can successfully synthesize proofs for 58% of the first 100/260 lemmas from the newly published Busy Beaver proofs.

## 1 Introduction

The general knowledge and reasoning abilities of frontier large language models (LLMs) makes them practical as a backbone for building agents able to interact with interactive theorem provers (ITP). These agents should iteratively build proofs with help from proof engine feedback. While previous work (e.g. Yang et al. [2023]) used a costly data collection procedure to finetune modestly sized language models, we believe that reasoning in natural language before outputting tactics will lead to better and more interpretable results. Recently, Thakur et al. [2024] showed promising preliminary results by using GPT-4 as an agent proposing tactics inside a backtracking search and using rich feedback from the proof environment.

In this work, we develop infrastructure to allow communication between a GPT-4o-based agent and the Coq proof environment [The Coq Development Team, 2024]. Our key idea is to rely on natural language as much as possible when generating proofs. Using natural language leverages the strength of LLMs, and allows us to use chain-of-thought [Wei et al., 2022] by asking for an informal mathematical proof before generating the formal proof, making it more intuitive and comprehensible compared to purely automatic formal techniques. Additionally, partial proofs expressed in natural language are easier for humans to understand, adapt, or reuse, allowing for greater flexibility and collaboration between machine-generated suggestions and human mathematicians.

We present the following contributions: 1) *Pétanque*: A new fast and lightweight environment to interact with the Coq theorem prover. 2) Two interactive proof protocols both leveraging natural language reasoning: tactic-by-tactic proof construction, and hierarchical proof templating. 3) We couple both protocols with standard search algorithms leveraging feedback from the ITP and using natural language to rerank proof candidates. 4) We evaluate this agent on a new dataset of textbook

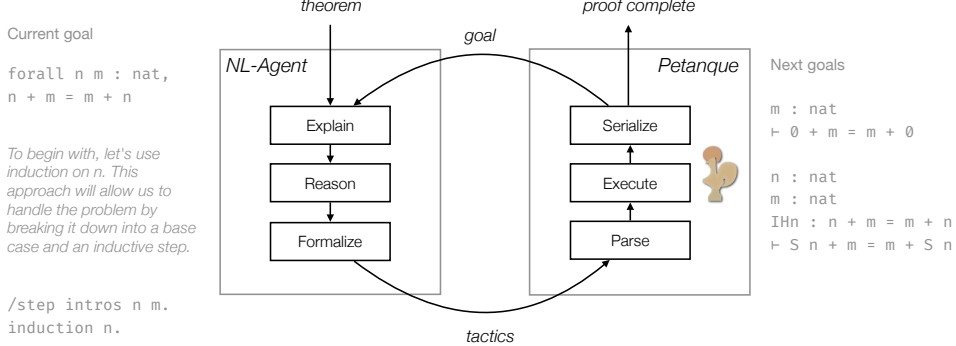

Figure 1: Tactic-by-tactic proof construction.

exercises and intermediate theorems from the recent Busy Beaver proof formalized in Coq of $BB(4) = 107$, [ccz181078, 2024]. NLIR is open source (https://github.com/llm4coq/nlir).

## 2   Pétanque: a lightweight interactive environment for Coq

A common difficulty when interacting with interactive proof assistants in the context of machine learning is inadequate tooling (see for example [Reichel et al.]). Following existing work [Gallego Arias et al., 2016, Gallego Arias, 2019, Yang and Deng, 2019, Sanchez-Stern et al., 2020], we have built a new environment for machine to machine interaction for the Coq proof assistant, particularly tailored for interactive, high-throughput, low-latency learning applications. Pétanque is based on Flèche [Gallego Arias, 2024], a new document manager for Coq. We extend Flèche by enabling Pétanque to access the Coq proof engine directly without requiring edits in the associated document. This makes our environment fast and lightweight. A Python interface provides easy access to the API. See Appendix B for more information on Flèche and Pétanque.

## 3   Proof interaction protocols

In this section, we present two approaches leveraging LLMs' ability to reason in natural language in order to find a formal proof with the help of a proof assistant. *Tactic-by-tactic proof construction* mimics the typical behavior of a standard Coq user: given the current goals, the agent generates one or several tactics that updates the goals and repeats this process until the proof is complete. By contrast, *hierarchical proof templating* tries to generate full proofs directly. Failed tactics are then replaced with *holes* to obtain a proof *template*. The agent repeats the process of filling each hole until the proof is complete. Our approach's originality is that although both protocols' inputs (goals) and outputs (tactics) are Coq code, the agent internally uses natural language as an intermediate representation to analyze the input and guide the code generation.

### 3.1   Tactic-by-tactic proof construction

An overview of the tactic-by-tactic proof construction agent is presented in Figure 1. Given a Coq theorem, the agent first uses natural language to describe the goal and explain how to continue the proof (chain-of-thought). The last step synthesizes the corresponding Coq tactics. For instance, in Figure 1, the goal is to prove that addition over natural numbers is commutative. The agent decides to try a proof by induction and correctly synthesizes a sequence of two tactics: **intros** n m. introduces two variables n and m of type nat (natural numbers), and **induction** n. starts an induction over n.

The tactics are sent to the Pétanque environment, which parses and executes each tactic to update the current goal. A textual representation of the new goal is then fed back to the agent to make further progress in the proof. If the execution returns an error, the current goal does not change, but we augment the prompt with the failed tactics and ask the LLM to try something else for the next attempt. For instance, in Figure 1, both tactics succeed and generate two new subgoals: the base case (for n=0, prove m + 0 = 0 + m) and the induction case (given the induction hypothesis

Figure 2: Hierarchical proof templating.

IHn$:$ n $+$ m $=$ m $+$ n, prove (n $+$ 1) $+$ m $=$ m $+$ (n $+$ 1) ). The textual representation of a goal uses the the symbol $\vdash$ to separate hypotheses from the conclusion, and S n denotes $n + 1$.

**Model Interface.** In early experiments, we observed that conversation-style reasoning often diverges: after a few rounds, the output makes very little sense, and the agent never recovers. Following [Yang et al., 2024] – and similarly to [Thakur et al., 2024] – we use a synthetic interface to summarize at each goal the global objective (initial theorem), the current goal (in the middle of a proof), and failed attempts to solve the same goal.

### 3.2 Hierarchical proof templating

An example execution of the hierarchical proof templating agent is presented in Figure 2. The agent pipeline is similar to the tactic-by-tactic method, but instead of focusing only on the next step, the agent generates a complete proof in natural language, before translating the proof in Coq syntax. For instance, in Figure 2, the agent uses the **inversion** tactics on the hypothesis H which generate two subgoals with a simpler hypothesis H0, and then tries to solve each subgoals using this H0 hypothesis.

The Pétanque environment then repairs the proof, replacing failed tactics by *holes* which admit and close the current subgoal, removing subsequent tactics until the focus moves to the next subgoal. Pétanque then checks that the resulting *template* is correct, i.e., assuming a valid proof for each holes, the proof is complete. A textual representation of each holes is then fed back to the agent which repeat the process to fill the holes one by one. For instance, in Figure 2, **apply** H0 fails on both subgoals. The agent then repeats the process for each holes, using focused fine-grain reasoning to prove the corresponding subgoal. The proof is complete when there are no more holes.

## 4 Proof search

We combine our interactive protocol with the classic beam search algorithm. Inspired by [Yao et al., 2023], we use the LLM to rank and sort the proposals at each step of the search. A simplified version of the code is presented on the right. At each step, agent.generate generates n_actions possible steps (tactics or proofs). Each step is then validated with petanque.step and the state of all the resulting candidates is stored. Then agent.sort calls the LLM to discuss, compare and finally rank and sort the candidates for the next step.

```python
def beam_search(n_steps, n_actions, beam_size):
  s = petanque.start(thm)
  beam = [s] # Initial state
  for step in range(n_steps):
    candidates = []
    for s in beam:
      # Try multiple actions for each state
      for a in agent.generate(s, n_actions):
        sa = petanque.step(s, a)
        if petanque.proof_finished(sa):
          return sa.proof  # Proof found!
        else:
          candidates = candidates + [sa]
    # Rank and sort candidates
    beam = agent.sort(candidates)[:beam_size]
  return None # No proof found
```

For the tactic-by-tactic agent, the state contains the current goal obtained by running all the previous steps from the initial goal, i.e., the theorem statement. At each step, `agent.generate` generates multiple possible tactics for the current goal. For each tactic proposed by the LLM, `petanque.step` executes the tactic to compute the updated state. If the tactic is invalid, we log the failure and the state is not modified.

For the template agent, the state contains a template, i.e. a proof with holes and a queue containing pointers to these holes and the associated goals. At each step, `agent.generate` generates multiple possible proofs for the first hole in the queue. For each proposed proof, `petanque.step` builds the corresponding sub-template. The updated state is obtained by replacing the current hole by the sub-template and adding the sub-templates holes to the end of the queue.

As a baseline, our naive search corresponds to a beam search with `n_action=1` and `beam_size=1` (in which case, the sorting step is useless).

## 5   Evaluation

**Logical Foundations exercises**: We extracted the exercises of *Logical Foundations* [Pierce et al., 2024], the first volume of the *Software Foundation* textbooks series that is widely used to introduce Coq. We extracted 179 exercices. Given the popularity of this textbook the risk of data leak is high. We filtered out 66 "easy" exercises that are solved with one shot prompting. This dataset thus comprises 113 exercises.

**BB$(4)$ lemmas**: To avoid data leak issues, we extracted the 260 lemmas from the recent proof of BB$(4) = 107$ [ccz181078, 2024]. The repository was created in April 2024, long after the knowledge cutoff date of the current version of GPT-4o (October 2023). To provide the necessary context for the proof, for each lemma we augment the prompt with all the preceding definitions and lemmas.

**Evaluation.**   The results are presented in the following table. We use Coq 8.19.2 and GPT-4o version 2024-05-13 for all the experiments.

| | *Logical Foundations* | | | | BB$(4)$ | |
| | tactics | | template | | template | |
| | naive | beam | naive | beam | naive | beam |
|---|---|---|---|---|---|---|
| % success | 30.1 | 46.0 | (13.3) 25.6 | (23.9) 38.9 | (21.0) 38.0 | (38.0) 58.0 |

For both agents, we set `n_actions=4` and `beam_size=3`, with `n_steps=30` for the tactics agent and `n_steps=10` for the template agent. While the tactics agent outperforms the template agent on the Logical Foundation benchmark, we observe that the template agent is significantly cheaper and faster than the tactics agent. By design the tactics agent requires much more interactions with the LLM to reach a full proof step by step.

To limit the costs of our experiments, we only run the template agent on the first 100 Lemmas of the BB$(4)$ benchmark. For the template agent, the gray numbers indicate the proportion of proofs that are correct at the first try (no holes). See Appendix A and Tables 1 and 2 for more details.

## 6   Related work and conclusion

**LLMs and theorem provers**   Automatic theorem-proving is a longstanding challenge in computer science [Newell et al., 1957]. Recent work has used neural models based on autoregressive language model that generate a proof tactic by tactic. Most works use finetuned LLMs [Polu and Sutskever, 2020, Han et al., 2021, Wu et al., 2022, Yang et al., 2023, First et al., 2023], trained on (goal, tactic) pairs obtained from intermediate steps of existing proofs. On the other hand, Lample et al. [2022] use online training, progressively collecting more data. Closest to our work, Thakur et al. [2024] build a tactic-by-tactic LLM agent based on GPT-4 and also use an interface to summarize past interactions. They, however, do not use proof repair or beam search. Also close to our work, Wang et al. [2024] use proof repair over hierarchical proofs in Isabelle, coupled with best-first search. Contrary to us, they use fine-tuned models and no chain-of-thought. Finally, Lin et al. [2024] propose a framework for

training language models to produce informal thoughts prior to each step of a proof, thereby boosting the model's theorem-proving capabilities.

**Reasoning in LLMs** This work is also related to recent investigations on the reasoning abilities of LLMs [Plaat et al., 2024]. Chain-of-Thought (CoT) prompting [Wei et al., 2022] was shown to improve LLM's answers; subsequent work found that these reasoning abilities could be elicited zero-shot [Kojima et al., 2022]. Further work interleaved CoT with decision-making [Yao et al., 2022], added search and complex control flow to reasoning [Chen et al., 2022, Yao et al., 2023, Besta et al., 2024], incorporated refinement and feedback [Madaan et al., 2024, Shinn et al., 2024], and learned to generate novel reasoning traces that proved beneficial for further training [Zelikman et al., 2022, 2024]. Like our work, many of these methods – especially the ones using search and refinement – make use of LLM-based scoring or ranking functions [Zheng et al., 2023].

**Conclusion** In this work, we have presented a new agent for building proofs leveraging chain of thought as an intermediate representation, and generating proofs by outputting step-by-step tactics or hierarchical proof templates. We couple this with beam search and natural language reranking and obtain good performance on a new evaluation set built with the help of our novel proof environment, *Pétanque*. Future work could investigate how one could use reinforcement learning to obtain better reasoning and performance with smaller models [OpenAI, 2024].

**Acknowledgements** We thank Cyril Cohen and Pierre Boutillier for many insightful discussions. We also thank Alex Sanchez-Stern for his feedback on early versions of Pétanque. This work is supported by the Inria Défi LLM4Code and the project ReaLiSe, Émergence Ville de Paris 2021-2025.

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

# A Detailed results

## A.1 Logical Foundations

For the template agent, the gray numbers indicate the proportion of proofs that are correct at the first try (no holes). We also report the average length of the generated proofs (number of tactics) and the size of the smallest and the biggest proofs. Details are presented in Table 1.

|  | tactics | | template | | |
| --- | --- | --- | --- | --- | --- |
|  | naive | beam | naive | beam | total |
| # success | 34 | 52 | (15) 29 | (27) 44 | 113 |
| % success | 30.1 | 46.0 | 25.6 | 38.9 | 100.0 |
| average proof length | 10.6 | 9.4 | 16.0 | 12.4 | |
| (min, max) proof length | (4, 31) | (4, 53) | (4, 55) | (4, 58) | |

## A.2 BB(4)

For each methods, we also report the original proof sizes (mean, min, and max) on the set of lemmas that were successfully proved. Details are presented in Table 2.

|  | template | | |
| --- | --- | --- | --- |
|  | naive | beam | total |
| # success | (21) 38 | (38) 58 | 100 |
| % success | 38.0 | 58.0 | 100.0 |
| average proof length | 13.7 | 15.4 | |
| original average proof length | 7.4 | 7.9 | |
| (min, max) proof length | (3, 38) | (3, 54) | |
| original (min, max) proof length | (2, 34) | (2, 34) | |

# B From Flèche to Pétanque

In this section we will describe Pétanque, a new environment for lightweight interaction with formal proof documents. Pétanque targets machine-learning applications such as reinforcement learning and other agent-based use cases, providing *zero-overhead*, *purely functional*[1] access to Coq's proof engine, along with some utilities to implement custom proof search routines.

**Flèche** Pétanque is built on top of Flèche [Gallego Arias, 2024], a new document manager for Coq. Flèche is both a formal document interpreter and a build system for Coq proof documents.

A schematic view of Flèche's behavior when the document is edited is presented in Figure 3. Flèche maintains an enriched representation of Coq proof documents, including the relevant Coq states associated to the interactive proofs and their dependencies. When an edit occurs, Flèche only invalidate the parts of the document that depend on that change, following standard incremental computing practices [Acar et al., 2005].

At any point, users can query Flèche for data about the document — for example information about the current proof obligations at a given point of the document — and Flèche will compute the requested information on-demand, as fast as possible.

**coq-lsp** Flèche's edit / query interface accommodates seamlessly the Language Server Protocol (LSP) protocol, the standard way to provide programming language support in modern Integrated Development Enviroments (IDEs). The LSP server coq-lsp[2] built on top of Flèche thus provides continuous real-time checking for Coq documents inside popular editors such as Emacs or VSCode.

---

[1]computations are treated as stateless functions, i.e., for equal inputs, we obtain equal outputs
[2]https://github.com/ejgallego/coq-lsp

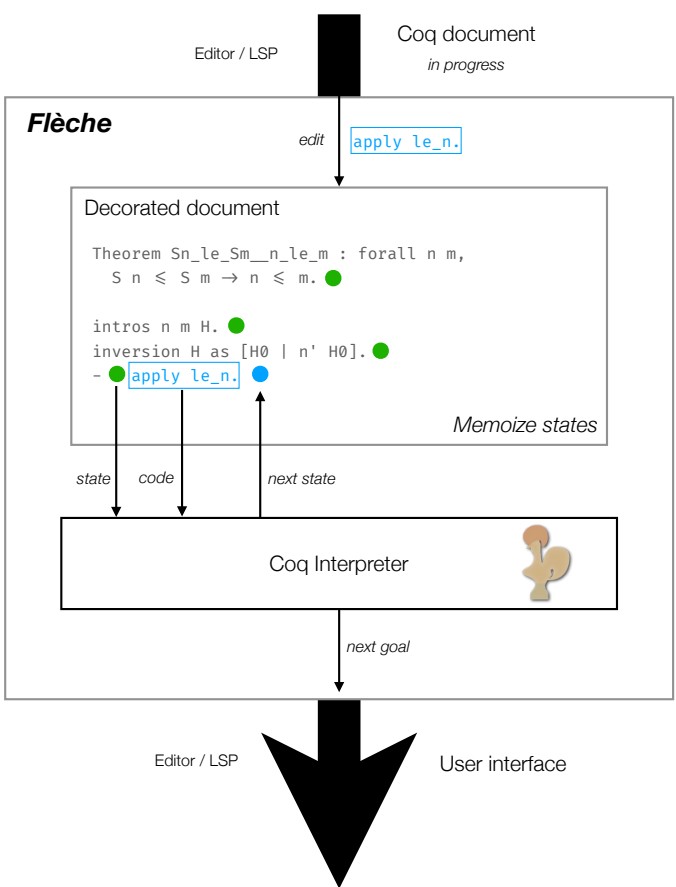

Figure 3: Flèche: a document manager for Coq. Flèche maintains a *decorated document* where each atom (definitions and proof steps) are associated with the Coq state (green dots). When an edit happens in the editor, Flèche retrieves the corresponding state, execute the code with the Coq interpreter, stores the new state (blue dot) in the decorated document, and returns the next goal that can be visualized in the editor. Communication with the editor relies on the LSP protocol.

```
class Pytanque:
  def start(self, file: str, thm: str) -> State
  def run_tac(self, state: State, tac: str) -> State
  def goals(self, state: State) -> List[Goal]
```

Figure 4: A simplified view of the pytanque API

**Pétanque**   Unfortunately the edit/query document model turns out to be too expensive for high-throughput, proof-search applications: while Flèche invalidation on edits is very efficient, the associated overhead starts to become a problem when the edit frequency is higher than a few times per second. Moreover, using IDE protocols such as LSP means that agents need to exchange message with the server multiple times per step, which again creates non-trivial overhead. To overcome the previous problems, Pétanque provides one-shot direct access to Coq's proof state and tactic engine.

A simplified view of the Pétanque API is presented in Figure 4. Using this API, agents can perform *speculative proof checking* without altering the original document.

The start methods initialize a proof session where the initial Coq proof state correspond to the theorem statement thm in the document file. Then, given a state, the run_tac method executes a tactic tac and return the resulting state if successful. The goals method can be used to retrieve a human readable version of the proof goals (e.g., as in Figures 1 and 2).

Table 1: Detailed results for the Logical Foundations benchmark.

| | tactics | | template | |
|---|---|---|---|---|
| | naive | beam | naive | beam |
| Basics:andb_true_elim2 | 12 | 9 | 10 | 10 |
| Basics:lower_letter_lowers | x | 7 | x | 8 |
| Basics:grade_lowered_once | 11 | 6 | 10 | 6 |
| Lists:eqblist_refl | x | x | x | x |
| Lists:count_member_nonzero | x | x | x | x |
| Lists:remove_does_not_increase_count | x | x | x | x |
| Lists:involution_injective | x | 8 | 7 | 7 |
| Lists:option_elim_hd | x | x | x | x |
| Lists:eqb_id_refl | x | 6 | 14 | 14 |
| Lists:update_eq | 19 | 6 | x | 9 |
| Lists:update_neq | 13 | 6 | 11 | 7 |
| Induction:add_comm | x | 10 | x | x |
| Induction:even_S | 12 | x | x | x |
| Induction:add_shuffle3 | 11 | 10 | x | x |
| Induction:mul_comm | x | 16 | x | x |
| Induction:plus_leb_compat_l | x | x | x | x |
| Induction:mult_plus_distr_r | x | x | x | x |
| Induction:mult_assoc | x | 11 | x | x |
| Induction:add_shuffle3' | 9 | 10 | x | x |
| Induction:bin_to_nat_pres_incr | x | x | x | x |
| Induction:nat_bin_nat | x | x | x | x |
| Induction:bin_nat_bin | x | x | x | x |
| Imp:optimize_0plus_b_sound | x | x | x | x |
| Imp:pup_to_2_ceval | x | x | x | x |
| Imp:loop_never_stops | x | x | x | x |
| Imp:no_whiles_eqv | x | x | x | x |
| Imp:execute_app | x | x | x | x |
| Imp:s_compile_correct | x | x | x | x |
| Imp:break_ignore | 4 | 4 | 4 | 4 |
| Imp:while_continue | 5 | 4 | 4 | 6 |
| Imp:while_stops_on_break | x | 4 | x | x |
| Imp:seq_continue | x | x | x | x |
| Imp:seq_stops_on_break | 4 | 5 | x | x |
| Imp:while_break_true | 4 | 4 | 4 | 4 |
| Imp:ceval_deterministic | x | x | 30 | 9 |
| IndProp:ev_double | 9 | 7 | 11 | 11 |
| IndProp:ev5_nonsense | 7 | 7 | x | x |
| IndProp:ev'_ev | x | x | x | x |
| IndProp:ev_plus_plus | x | x | x | x |
| IndProp:total_relation_is_total | x | x | x | x |
| IndProp:empty_relation_is_empty | 5 | 5 | 7 | 8 |
| IndProp:O_le_n | 4 | 4 | 10 | 10 |
| IndProp:Sn_le_Sm__n_le_m | 16 | 5 | 9 | 9 |
| IndProp:lt_ge_cases | x | x | x | x |
| IndProp:le_plus_l | x | 6 | 11 | 11 |
| IndProp:plus_le | x | x | x | x |
| IndProp:add_le_cases | x | x | x | x |
| IndProp:plus_le_compat_r | x | 14 | x | x |
| IndProp:le_plus_trans | x | 15 | x | x |
| IndProp:n_lt_m__n_le_m | x | 6 | 7 | 9 |
| IndProp:plus_lt | x | x | x | x |
| IndProp:leb_complete | x | x | x | 23 |
| IndProp:leb_correct | x | x | x | x |
| IndProp:leb_true_trans | 12 | 11 | x | 11 |
| IndProp:R_equiv_fR | x | x | x | x |
| IndProp:subseq_refl | x | x | x | x |
| IndProp:subseq_app | 8 | 4 | 4 | 4 |
| IndProp:subseq_trans | 4 | 4 | x | 6 |
| IndProp:reflect_iff | 11 | 12 | 20 | 18 |
| IndProp:eqbP_practice | x | x | x | x |
| IndProp:merge_filter | 19 | 4 | 29 | 4 |
| IndProp:pal_app_rev | x | x | x | x |
| IndProp:pal_rev | 4 | 4 | 4 | 4 |
| IndProp:palindrome_converse | x | x | x | x |
| IndProp:pigeonhole_principle | x | x | x | x |
| IndProp:regex_match_correct | x | x | x | x |
| Poly:rev_involutive | 14 | 9 | 12 | 12 |
| Poly:map_rev | x | x | x | x |
| Poly:uncurry_curry | x | x | x | x |
| Poly:curry_uncurry | x | x | x | x |
| ImpCEvalFun:ceval__ceval_step | x | x | x | x |
| Logic:leb_plus_exists | x | x | x | x |
| Logic:In_map_iff | 31 | 28 | x | 46 |
| Logic:In_app_iff | x | x | 55 | x |
| Logic:All_In | x | x | x | x |
| Logic:combine_odd_even_intro | x | x | x | x |
| Logic:combine_odd_even_elim_odd | x | x | x | x |
| Logic:combine_odd_even_elim_even | x | x | x | x |
| Logic:eqb_neq | x | 15 | x | x |
| Logic:eqb_list_true_iff | x | x | x | x |
| Logic:forallb_true_iff | x | x | x | x |
| Logic:tr_rev_correct | x | x | x | x |
| Logic:excluded_middle_irrefutable | x | 16 | x | 16 |
| Rel:total_relation_not_partial_function | x | x | x | x |
| Rel:lt_trans' | 18 | 6 | x | 4 |
| Rel:lt_trans'' | 18 | 9 | x | 12 |
| Rel:le_S_n | 7 | 5 | x | 9 |
| Rel:le_not_symmetric | x | 7 | x | 7 |
| Rel:le_antisymmetric | 7 | 9 | x | x |
| Rel:le_step | x | x | x | x |
| Rel:rtc_rsc_coincide | x | x | x | 30 |
| IndPrinciples:booltree_ind_type_correct | x | x | x | x |
| IndPrinciples:Toy_correct | x | x | x | x |
| IndPrinciples:reflect_involution | x | x | x | x |
| Maps:t_update_neq | x | 12 | 10 | 14 |
| Maps:t_update_permute | x | x | x | x |
| Tactics:rev_exercise1 | 9 | 7 | 17 | 15 |
| Tactics:eqb_true | x | x | x | x |
| Tactics:plus_n_n_injective | x | x | 34 | x |
| Tactics:combine_split | x | x | 21 | 20 |
| Tactics:bool_fn_applied_thrice | 21 | 16 | 35 | x |
| Tactics:eqb_sym | x | x | x | 17 |
| Tactics:eqb_trans | 10 | x | x | x |
| Tactics:split_combine | x | x | x | x |
| Tactics:existsb_existsb' | x | x | x | x |
| ProofObjects:ev_8 | 7 | 7 | 7 | 7 |
| ProofObjects:pe_implies_pi | x | 12 | x | 11 |
| AltAuto:ev100 | x | 53 | 58 | 55 |
| AltAuto:andb3_exchange | x | 4 | x | 4 |
| AltAuto:andb_true_elim2 | 4 | 6 | 10 | 10 |
| AltAuto:andb3_exchange' | x | 12 | x | 23 |
| AltAuto:nor_comm' | 12 | 10 | x | 10 |
| AltAuto:nor_not' | x | 11 | x | 10 |

Table 2: Detailed results for the $\mathrm{BB}(4)$ benchmark.

| | orig. | naive | beam |
|---|---|---|---|
| ffx_eq_x_inj | 10 | 7 | 7 |
| enc_v1_eq | 6 | x | x |
| enc_pair_inj | 12 | x | x |
| enc_list_inj | 16 | x | x |
| andb_shortcut_spec | 3 | 7 | 9 |
| orb_shortcut_spec | 3 | 9 | 7 |
| set_ins_spec | 33 | x | x |
| empty_set_WF | 10 | 19 | 16 |
| pop_back_len | 8 | x | 20 |
| pop_back__nth_error | 15 | x | 54 |
| list_eq__nth_error | 34 | 37 | 44 |
| pop_back'__push_back | 6 | x | x |
| St_enc_inj | 2 | 5 | 4 |
| St_eqb_spec | 3 | 3 | 4 |
| Sigma_eqb_spec | 3 | x | x |
| Sigma_enc_inj | 2 | x | x |
| listSigma_inj | 12 | 38 | 23 |
| map_inj | 9 | 29 | 29 |
| listT_enc_inj | 7 | 6 | 6 |
| Dir_eqb_spec | 3 | 11 | 3 |
| St_list_spec | 4 | x | 12 |
| Sigma_list_spec | 4 | 13 | 8 |
| Dir_list_spec | 4 | 13 | 13 |
| forallb_St_spec | 9 | x | 14 |
| forallb_Sigma_spec | 9 | 18 | 17 |
| forallb_Dir_spec | 9 | x | 13 |
| Steps_trans | 9 | x | x |
| Steps_unique | 11 | x | 19 |
| Steps_NonHalt | 22 | x | x |
| HaltsAt_unique | 16 | x | x |
| NonHalt_iff | 27 | x | x |
| LE_step | 10 | x | 14 |
| LE_Steps | 10 | 13 | 12 |
| LE_NonHalts | 8 | x | x |
| HaltTimeUpperBound_LE_NonHalt | 7 | x | x |
| LE_HaltsAtES_1 | 11 | x | x |
| LE_HaltsAtES_2 | 14 | x | x |
| HaltTimeUpperBound_LE_Halt | 15 | x | x |
| St_swap_swap | 12 | x | x |
| Trans_swap_swap | 7 | 8 | 8 |
| option_Trans_swap_swap | 7 | 10 | 10 |
| TM_swap_swap | 8 | x | 15 |
| ExecState_swap_swap | 7 | 6 | 6 |
| step_swap | 18 | x | 48 |
| step_halt_swap | 10 | x | 39 |
| Steps_swap | 27 | x | x |
| LE_swap_0 | 7 | x | 23 |
| LE_swap | 9 | x | x |
| InitES_swap | 8 | x | 15 |
| HaltsAt_swap_0 | 15 | x | 17 |

| | orig. | naive | beam |
|---|---|---|---|
| HaltsAt_swap | 9 | 31 | 30 |
| HaltTimeUpperBound_LE_swap | 10 | x | x |
| HaltTimeUpperBound_LE_swap_InitES | 5 | x | x |
| Trans_rev_rev | 7 | 6 | 8 |
| option_Trans_rev_rev | 8 | 11 | 10 |
| TM_rev_rev | 7 | 8 | 11 |
| Tape_rev_rev | 7 | 12 | 9 |
| ExecState_rev_rev | 7 | 6 | 6 |
| fext_inv | 3 | 5 | 5 |
| step_rev | 44 | x | x |
| step_halt_rev | 11 | x | x |
| Steps_rev | 27 | x | x |
| LE_rev_0 | 7 | 19 | 19 |
| LE_rev | 9 | x | x |
| InitES_rev | 3 | 8 | 6 |
| HaltsAt_rev_0 | 15 | 20 | 18 |
| HaltsAt_rev | 9 | x | x |
| HaltTimeUpperBound_LE_rev | 10 | x | x |
| HaltTimeUpperBound_LE_rev_InitES | 5 | x | x |
| Trans_swap_id | 10 | x | x |
| isUnusedState_spec | 58 | x | x |
| step_UnusedState | 11 | 13 | 17 |
| Steps_UnusedState | 15 | x | x |
| HaltTimeUpperBound_LE_HaltsAtES_UnusedState | 68 | x | x |
| TM0_LE | 7 | x | x |
| UnusedState_TM0 | 10 | 12 | 21 |
| UnusedState_dec | 4 | x | 12 |
| HaltTimeUpperBound_LE_HaltAtES_MergeUnusedState | 31 | x | x |
| St_to_nat_inj | 4 | 5 | 5 |
| St_suc_le | 4 | x | 3 |
| St_suc_eq | 5 | x | 14 |
| St_suc_neq | 3 | 17 | 8 |
| HaltTimeUpperBound_LE_HaltAtES_UnusedState_ptr | 21 | x | x |
| HaltsAtES_Trans | 9 | x | 25 |
| UnusedState_upd | 68 | x | x |
| UnusedState_ptr_upd | 97 | x | x |
| isHaltTrans_0 | 3 | 21 | 18 |
| CountHaltTrans_upd | 7 | x | x |
| CountHaltTrans_0_NonHalt | 21 | x | x |
| Trans_list_spec | 6 | x | 8 |
| St_leb_spec | 13 | x | 10 |
| TM_simplify_spec | 6 | 9 | 7 |
| TM_upd'_spec | 5 | 9 | 8 |
| nat_eqb_spec | 3 | 11 | 11 |
| TNF_Node_expand_spec | 64 | x | x |
| TNF_Node_NonHalt | 6 | x | 9 |
| HaltDecider_cons_spec | 7 | 16 | 39 |
| SearchQueue_upd_spec | 74 | x | x |
| SearchQueue_upd_bfs_spec | 30 | x | x |
| SearchQueue_reset_spec | 13 | 29 | 26 |

