# OpenReview forum: "NLIR: Natural Language Intermediate Representation for Mechanized Theorem Proving"
_NeurIPS.cc/2024/Workshop/MATH-AI — MATH-AI 24_

### Official Review · Reviewer_DamW · 2024-09-30
**The paper introduces an innovative approach by leveraging large language models (LLMs) like GPT-4 as intermediaries for mechanized theorem proving using natural language, which enhances interpretability and collaboration. The introduction of the Pétanque environment and interactive proof protocols shows strong potential, with promising evaluation results on Logical Foundations and Busy Beaver datasets. However, the paper could benefit from deeper theoretical exploration, broader dataset evaluation, and a more thorough comparison with alternative systems. Overall, it is a strong contribution to the field with room for further development.**

**Rating:** 9
**Confidence:** 4

**Review:**

# Review for "NLIR: Natural Language Intermediate Representation for Mechanized Theorem Proving"

## Strengths

### 1. Innovative Approach
The paper introduces a novel approach to leveraging large language models (LLMs) such as GPT-4 to serve as intermediaries in mechanized theorem proving. By employing natural language as an intermediate layer, it simplifies interactions between users and theorem provers. This technique is a valuable contribution that brings interpretability and flexibility to the process of formal proof generation.

### 2. Lightweight Environment (Pétanque)
The introduction of Pétanque, a fast and lightweight environment for interacting with the Coq theorem prover, is highly commendable. The environment is tailored to high-throughput, low-latency applications, making it a valuable tool for both machine-machine and human-machine theorem proving interactions.

### 3. Interactive Proof Protocols
The authors present two proof construction protocols: *tactic-by-tactic* and *hierarchical proof templating*, both of which utilize natural language reasoning. These protocols allow for more interpretable proof generation compared to purely formal methods and promote greater collaboration between machines and human mathematicians.

### 4. Comprehensive Evaluation
The paper evaluates its approach on two datasets: Logical Foundations exercises and Busy Beaver proofs. The evaluation metrics are clearly presented, showing that the method achieves a 46% success rate on Logical Foundations and a 50% success rate on Busy Beaver proofs, which demonstrates the practical effectiveness of the approach.

## Weaknesses

### 1. Limited Theoretical Explanation
While the paper presents impressive empirical results, it lacks a deeper theoretical explanation of why natural language serves as an effective intermediate representation in formal theorem proving. More discussion on the theoretical underpinnings and benefits of this method would provide stronger validation.

### 2. Narrow Dataset Evaluation
The evaluation focuses on two specific datasets. Expanding the scope to include a broader variety of theorems, including more complex or abstract ones, would provide a better understanding of how generalizable the proposed approach is across different domains.

### 3. Over-reliance on LLMs
The paper does not thoroughly address the limitations of relying heavily on LLMs like GPT-4, especially for scaling to more complex mathematical problems. It would be beneficial to explore potential limitations or challenges that arise when applying the approach to larger or more complex proof tasks.

### 4. Lack of Comparative Analysis
While the paper references prior works, a more detailed comparative analysis with alternative systems or approaches (e.g., symbolic AI theorem proving) would provide a clearer context of how this approach stacks up against state-of-the-art methods.

## Suggestions for Improvement

- **Expand Theoretical Discussion**: A deeper discussion on why natural language works as an effective intermediary between humans and formal proof systems could improve the paper's impact and clarify its contributions.

- **Broader Dataset Evaluation**: Expanding the evaluation beyond Logical Foundations and Busy Beaver proofs to other complex theorems would showcase the flexibility and robustness of the approach.

- **Address LLM Limitations**: A discussion of potential pitfalls, such as the scalability of GPT-4 to more complex or abstract mathematical domains, would strengthen the validity of the proposed method.

- **Incorporate Comparative Studies**: Direct comparisons with alternative theorem proving systems or strategies could offer valuable insights into the strengths and weaknesses of this method relative to other state-of-the-art approaches.

## Conclusion

The paper presents a promising and innovative approach to theorem proving by integrating natural language and LLMs into formal systems like Coq. The introduction of Pétanque and the two proof protocols provide a foundation for future research in this space. With further development, particularly around theoretical exploration and broadening the scope of evaluation, this work has the potential to become highly impactful in the field of mechanized theorem proving.

---

### Official Review · Reviewer_1uCN · 2024-10-02
**Interacting between NL and Coq for theorem proving**

**Rating:** 4
**Confidence:** 4

**Review:**

Overall, I think this paper presents a useful tool for Coq theorem proving, and for bridging NL and Coq in this task. The novelty and significance of this work convinces me less. The idea of agent-based approaches, and more concretely, letting LLMs reason in informal languages via having access to the proof checking environment is not new, and has been realized already in other proof assistants (e.g. Lean). So is the idea of building Python interfaces. Below are some other specific comments:

1. Since Pétanque is directly based on Flèche, I was hoping for a more detailed and straightforward discussion of which things were done by which work.

2. When seeing failure proofs, the idea of leaving holes in it for further proving instead of just discarding it sounds interesting. However, I feel this is based on the assumption that the failure proofs at least have a right overall structure and direction, which may not always be the case. In cases where the generated proofs are just not along the right line, the hole (in some sense the subgoals broken down) may be bad subgoals at the first place, limiting the chance of finishing the proof subsequently by focusing on such holes.

3. The heuristic in the beam search is designed here as a LLM ranking. There are other works which simply use LLMs' likelihoods as the heuristic. It would be great to have a comparison between these two approaches.

4. I like how the authors make a successful effort to avoid data leakage in their evaluation. If they wish to continue this work and expand into a full conference paper, I would also recommend expanding the scope of experiments.

---

### Official Review · Reviewer_8Ri1 · 2024-10-08
**The paper is overall well-written and the main ideas are clearly explained.**

**Rating:** 5
**Confidence:** 2

**Review:**

Paper Summary:

This paper introduces a gpt4 math agent for the Coq prover, using either step-by-step interactive tactic generation and hierarchical proofs, as well as search. The authors evaluate Logical Foundations and Busy Beaver proof using a new environment.

Weakness and strengths:

(+) The paper is well-structured and presents the problem, methodology, and findings.

(+) This paper studies an interesting and well-motivated research topic: Formal theorem proving using LLM techniques.

(-) It would be beneficial to add more benchmarks in the experiments to evaluate the performance of the proposed method.

---

### Official Review · Reviewer_pnLu · 2024-10-09
**Review of NLIR**

**Rating:** 9
**Confidence:** 3

**Review:**

Summary：
This paper presented a new agent Pétanque for building proofs leveraging chain of thought as an intermediate representation, which obtain good performance on a new evaluation set.

Strength:
1. NL as general interface for reasoning about formal proofs is natural and important.
2. Based on this insight, authors design a lightweight and powerful proof tool Pétanque.
3. The proposed method show the outstanding performance and importance for theorm proof.

Weakness:
NL as an intermediate representation for reasoning actually have been proposed by [1] in theorm proof. Authors maybe should disscuss the difference.

[1] Lean-STaR: Learning to Interleave Thinking and Proving

---

### Decision · Program_Chairs · 2024-10-09

**Decision:**

Accept

**Comment:**

The reviewers are mixed on this work. While the work has limited novelty and lacks a comparison with related approaches, it presents an interesting system for theorem proving in Coq. We suggest the authors consider the comments mentioned in the reviews when submitting the camera-ready version.